# Planning and Organization of the COVID-19 Vaccination Campaign: An Overview of Eight European Countries

**DOI:** 10.3390/vaccines10101631

**Published:** 2022-09-28

**Authors:** Chiara Cadeddu, Aldo Rosano, Leonardo Villani, Giovanni Battista Coiante, Ilaria Minicucci, Domenico Pascucci, Chiara de Waure

**Affiliations:** 1Section of Hygiene, Department of Life Sciences and Public Health, Università Cattolica del Sacro Cuore, 00168 Rome, Italy; 2National Institute for the Analysis of Public Policy—INAPP, 00198 Rome, Italy; 3Department of Medicine and Surgery, University of Perugia, 06132 Perugia, Italy

**Keywords:** COVID-19 vaccines, immunization programs, vaccination coverage

## Abstract

The initial progress of the COVID-19 vaccination campaign worldwide depended on several aspects, including programmatic/practical issues. This paper focused on the planning and organization of COVID-19 vaccination campaigns in eight European countries (Sweden, Denmark, Romania, Hungary, Italy, Spain, Germany, and France), from the launch to August 2021. Information on the planning of the vaccination campaign (release and update of a national immunization plan, types of vaccines being used and their limitations/suspensions) and its organization (vaccination target groups, possibility of citizens’ choice, vaccination workforce and settings, vaccines procurement) were obtained through desk research of international and national reports, plans, and websites. Eventually, data on vaccination coverage were drawn from Our world in data and analyzed through join point regression. The eight countries showed differences in groups prioritization, limitations/suspensions of use of specific vaccines, citizens’ possibility to choose vaccines, and vaccination workforce involved. These issues could have contributed to the different progress towards high levels of vaccination coverage. In respect to vaccination coverage, Romania reached much lower levels than other countries. Further comparative research is needed in order to identify best practices in vaccination campaign that could be useful for the next phases of the COVID-19 pandemic, and be better prepared for future potential pandemic.

## 1. Introduction

As of July 22nd 2022, more than 565 million confirmed cases of COVID-19 and 6.3 million related deaths were reported to the World Health Organization (WHO) [1]. Non-Pharmaceutical Public Health Interventions (NPHIs), namely stay-at-home orders, social distancing, travel restriction, and lockdown, were adopted from the start of the pandemic and have proved to be effective [2,3], albeit deeply impacting on both countries’ economy and people’s life. Because of that, efforts have been immediately made to develop safe and effective vaccines to control the pandemic [4]. The first vaccines received conditional marketing authorization in less than a year and European Union (EU) countries started their vaccination campaigns in the last days of December 2020. The launch of the massive vaccination campaign across EU countries was made possible thanks to the European Commission (EC)’s commitment in securing access to safe, efficient, and high quality COVID-19 vaccines for all EU citizens. Nevertheless, the same EC has from the start drawn the attention on the successful deployment and uptake of vaccines identifying some fundamental pillars, namely the infrastructure, the workforce, the communication, the information system, and the monitoring of vaccination campaigns [5]. It is well known that vaccination uptake is influenced by several aspects, among them the availability of vaccines, which has changed over time during the pandemic, and subjective aspects that can influence people’s acceptability [6] and that are related to confidence, constraints, complacency, calculation, and collective responsibility [7]. Besides these aspects, others could also impact on vaccination uptake, namely, accessibility, accommodation, and affordability [6]. These three aspects refer to where and how the service is provided. With this respect, it needs to be said that conventional health care sites did immediately appear as insufficient to reach a high vaccine uptake on a timely basis [8]. Indeed, mass vaccination centers have been created to guarantee thousands of daily administrations [9]. At the same time, in order to reach an equitable vaccine distribution and access, especially in medically unserved rural areas, many countries have involved pharmacies in the vaccination campaign [10]. Public health authorities have also offered drive-in or mobile clinic services to populations living in rural areas or neighborhoods [11]. Eventually, many companies have used mobile clinics or existing occupational health clinics to vaccinate their employees and family members based on the evidence of the success of these initiatives in increasing vaccination uptake [12].

Up to now, the evidence on the factors influencing COVID-19 vaccine uptake has mostly addressed vaccine hesitancy and its determinants [13,14,15,16,17,18]. Nonetheless, among the reasons for vaccine hesitancy, logistic/practical issues have also been identified. This is not unexpected as, for some time, the WHO has recognized the role of programmatic/practical issues in both the implementation and the uptake of vaccines [19,20].

The planning and organization of a vaccination campaign is indeed an aspect to bear in mind and evaluate to increase vaccine uptake, especially during a mass pandemic vaccination campaign [21]. This paper aims to make an overview of aspects related to the planning and organization of COVID-19 vaccination campaigns in its initial eight months in a group of EU countries from different subregions of Europe. The secondary objective was also to speculate on the relationship between those aspects and the trend of vaccination coverage.

## 2. Materials and Methods

The work was based on a desk research, aiming to collect data on the COVID-19 vaccination campaign in the selected countries, and the collection and re-elaboration of COVID-19 epidemiological data and vaccination coverage until 31st August 2021. This date was chosen as it was the milestone of fully vaccinating 70% of the European adult population, set by the European Union [22], and was also the period that just preceded the authorization for the booster dose, given in autumn 2021 [23], that changed the framework, coverage, and efficacy issues related to variants and the considerations about management.

### 2.1. COVID-19 Vaccination Campaigns

We obtained information about vaccination policies from international and national sources. In particular, we consulted technical reports and the COVID-19 Vaccine Tracker of the European Center for Disease Prevention and Control (ECDC), the national COVID-19 national immunization plans (NIPs), where available, reports from national institutes, scientific literature, and governmental and institutional websites. Appendix A reports the sources used for each country.

We considered eight European countries belonging to different geographical areas of Europe: north (Sweden and Denmark), east (Romania and Hungary), central-west (Germany, and France) and south (Italy, and Spain). The choice of countries was opportunistic but aimed to also ensure the representativeness of the different health systems across Europe (both decentralized, tax-funded, and centralized social health insurance-based systems). Thus, in order to describe the vaccination campaign of each selected country, we considered two dimensions related to COVID-19 vaccination policies: type of vaccines used; and vaccination organization. In particular, with regard to the first dimension, we searched for information about the authorized vaccines (i.e., Spikevax, Comirnaty, Vaxzevria, Jansenn, Sputnik V, Sinopharm, CanSino, Covishield) and their indications for administration (e.g., age limit), limitations for administration (e.g., specific age groups), and possible suspension. 

Considering the vaccination campaign organization, we investigated the possibility to choose the vaccine by: citizens (yes, no, partly); the criteria for vaccination administration (priority for specific population groups); the vaccination workforce (medical doctors, retired medical doctors, nurses, pharmacists, other professionals); and the vaccination settings (hospitals, general practitioners (GPs)’ ambulatories, hubs, pharmacies, drive-thru units, mobile units, workplace). In addition, we sought the level of organization of the vaccination campaign (centralized, decentralized, mixed) and the distribution criteria on the territory (by population or by risk level). Finally, we searched for the availability of COVID-19 NIPs.

### 2.2. COVID-19 Epidemiology and Vaccination Coverage

For each country under study, data on incidence and mortality for COVID-19 and vaccination coverage concerning the first dose and full vaccination (intended as second dose) were drawn from the repository “Our world in data” [24], which uses official numbers from governments and health authorities. The incidence was calculated as the number of new cases of COVID-19 for every 1,000 people. Mortality rate was calculated as the number of new confirmed deaths for every 10,000 people. These indicators were calculated month by month from January 2021 to August 2021. 

Vaccination coverage was calculated as the proportion of eligible persons fully vaccinated. 

To identify potential changes in full vaccination coverage trends and the relationship between the vaccination coverage (response variable), and the time period (predictor), a join point regression model was estimated for every country by using the Join Point Regression (JPR) Program, Version 4.5.0.1 (Statistical Research and Applications Branch, National Cancer Institute). The null hypothesis of absence of changes in trend was tested using a maximum of two changes in slope with an overall significance level of 0.05. In brief, the JPR method assumes that data can be divided into subsets, each with their own unique linear trend and identifies the period (lasting 15 days each in our analysis) when a trend change is produced, calculates the percentage change (PC) in rates between trend-change points, and also estimates the average percentage change (APC) in the whole period studied. In case the data of the days 1 or 15 were not available, the available data in the closest day (the day(s) before or the day(s) after) were considered. As the full vaccination coverage, inevitably, showed percentages different from zero only starting from February 2021, the JPR considered the period from 1st of February 2021 to 1st of September 2021

## 3. Results

### 3.1. COVID-19 Vaccination Campaigns

The approval of the first vaccine against COVID-19 (Comirnaty) took place in the last 10 days of December 2020, in each country. Nevertheless, only two countries had already issued a COVID-19 NIP in the middle of the year (June—Germany, and August—Sweden). The remaining countries released a NIP at the end of the year. NIPs were released with regular updates (Sweden, Romania, Spain) or as a single version (Denmark, France, Germany, and Italy, with only one update following a governmental change), whereas for Hungary only a partial plan was available.

After the approval of other COVID-19 vaccines and their mass use, countries faced the problem of adverse events not reported in the registrational trials. All countries considered in the study, except Hungary, temporarily suspended the use of Vaxzevria after the occurrence of rare cases of blood clots.

Romania and Denmark were the first countries to do this on 11th March. Other countries followed on 15th March (France, Germany, Italy, and Spain) and 16th March (Sweden). The use of Vaxzevria was then soon resumed on 19th March in Romania, France, Germany, and Italy, while Spain and Sweden resumed it later, on 24th and 25th March, respectively. In contrast, Denmark went for its definite suspension. It should be observed that the resumption of vaccination with Vaxzevria came with the introduction of restrictions to its use based on age (use allowed in people ≥ 55 years old in France, ≥60 years old in Germany, Italy, Spain, and ≥65 years old in Sweden). 

#### Organizational Aspects of COVID-19 Vaccination Campaigns

Organizational aspects widely varied among the considered countries. 

In terms of the possibility for citizens to choose, or not, which type of vaccine to receive, only in Romania was a completely free choice offered, while in three countries (Denmark, France, Italy) at least a partial possibility of choice (e.g., choice limited between the two viral vector vaccines, choice related to the site that uses the preferred vaccine, and choice allowed only in some regions of the country) was available. In Sweden, Spain, and Germany, there was no option for choice, even if, in the last two countries, a citizen could refuse the vaccine declared at the time of the administration and book a new appointment afterwards. No information about this issue was found for Hungary.

According to priority criteria, homecare residents together with elderly and vulnerable groups were the first categories to receive COVID-19 vaccination in all countries. Other priority groups were heterogeneously identified: homecare residents’ caregivers; healthcare workers and all the other individuals at last (Sweden, Denmark); in a first phase homecare residents’ staff and caregivers over 50 years, and healthcare workers over 50 years; in a second phase, all healthcare workers, with vaccination becoming mandatory in September for healthcare workers and workers in long-term care facilities—LCTFs (France); adults (18–59 years) with comorbidities, people with disabilities, key workers (healthcare workers, personnel in LTCFs) with vaccination becoming mandatory from 31 March 2021, for healthcare workers and workers in LCTFs (Hungary); healthcare and social workers and then the whole population (Romania); healthcare workers, LTCFs staff, school and university staff (18–55 years) and security forces (18–55 years) with vaccination becoming mandatory from 1 April 2021 for healthcare workers and workers in LCTFs (Italy); LTCFs staff, healthcare workers and social-health personnel, prison workers, non-institutionalized third-degree dependents, security forces and educational staff and then people by age group (>60, >50, >40 years) (Spain).

In Germany, the priority started with just two groups of vulnerable people at the end of December 2020 and was extended in February 2021 to all the vulnerable people and, finally, in June, to the whole population.

In respect to the elderly, priority groups were also defined. In Italy, in February 2021, only the elderly aged 80 years or older were targets of the immunization, while in the following months vaccination was progressively extended backwards by decades per month (i.e., seventies, sixties, etc.).

In relation to the vaccination workforce, only in Hungary and Romania were medical doctors and nurses the only categories engaged, while in all other countries other healthcare professionals were also employed, including pharmacists in some cases. In Romania and in Italy, retired medical doctors were also called back to work.

Settings equipped for COVID-19 vaccination were mainly healthcare settings, such as hospitals, healthcare centers, GPs surgeries, and other settings such as dedicated hubs, drive-thru units, pharmacies, and mobile units. In Germany, workplaces were also employed.

The management of COVID-19 vaccination differed among countries: it was centralized in Denmark, France (with some regional activities), Hungary, and Romania; decentralized in Sweden and Germany (in each land); and mixed in Italy (centralized for operating standards and decentralized for localization, coordination, and control of vaccination activities) and Spain (centralized for vaccines’ distribution and decentralized for the supply of equipment and resources for administration).

The vaccine distribution criteria were organized mainly by risk level (by stages or by prioritized risk groups) or mixed, starting by risk level and becoming by population in the second phases of the campaign.

### 3.2. COVID-19 Epidemiology and Vaccination Coverage

The incidence of new cases and mortality for COVID-19 showed different patterns: peaks were observed in the months of March and April 2021 in France, Italy, Hungary, and Sweden; and in July and August in Spain. Summer months were characterized by low incidence in most of the studied countries. The highest rates of mortality were observed in Hungary and Romania (Figure 1).

All countries reached 40% of full COVID-19 vaccination coverage in July 2021, with the exception of Romania and Sweden. In August 2021, the highest level of people fully vaccinated was reached by Spain and Denmark (71%); a high level (around 60%) was also reached, in decreasing order, in Italy, Germany, and France. Hungary and Romania saw a slowdown in vaccination in the summer, while France and Spain had been able to increase the coverage by more than 20%. Coverage in Romania was constantly much lower than other countries; in August only around one fourth of eligible people were fully vaccinated.

Based on the join point regression analysis, it has been possible to identify three different phases in three (Germany, Hungary, and Spain) out of the eight countries: the start of the vaccination campaign with a fast increase in vaccination coverage (February–March); a second phase with a moderate increase in vaccination coverage (April–May); and a last one with a downturn in the increase of vaccination coverage (June–August). In these countries, two join points were indeed observed: the first around March, for Germany and Romania, and April for Hungary; the second in May/June for all three countries. Denmark, France, Italy, Spain, and Sweden were characterized by only two phases: a steady rapid increase until April/May; and a following reduction. In the upward phase, the PC ranged from 38.9% in Denmark to 365.1% in France. In the last phase, the PC ranged from 2.5% in Romania to 29.9% in Sweden (Table 1). The APC for the overall study period ranged from 28.7% in Romania to 57.0% in France.

## 4. Discussion

Our study provided details about the planning and organization of the COVID-19 vaccination campaign at its beginning in a few selected European countries, which can be considered key elements to explain the observed differences in the progress toward reaching high vaccination coverage. The latter has been the focus of both health policy makers and health professionals worldwide. It is not by chance that much literature has investigated people’s attitudes and behaviors toward COVID-19 vaccination to identify the reasons behind vaccine hesitancy. Nevertheless, beyond motivation, practical issues are also important in determining vaccination uptake and for the success of a vaccination campaign, as Israel’s experience has shown [25]. Our overview identified different approaches (i.e., centralized/regional/mixed) adopted to procure vaccines across the considered countries, that mainly resembled the organization of the countries’ health systems. An interesting element is that in Romania and Hungary, which had the lowest vaccination coverage rates compared to the other considered countries, the healthcare workforce involved in the vaccination campaign was limited to doctors and nurses. Furthermore, Romania allowed the free choice of vaccine, and this may have slowed the progression of the campaign, instead of being a positive factor.

By the end of August, 181.5 million persons were fully vaccinated in the eight countries selected for this study, representing 60% of the population. The highest level of people fully vaccinated was reached in Denmark and Spain, but a high level was also achieved, in descending order, in Italy, Germany, France, and Sweden. Hungary and Romania showed, in contrast, lower levels of vaccination, with less than one third of fully vaccinated people in Romania.

The vaccination campaign started simultaneously in all EU countries on 27th December 2020. The COVID-19 vaccine rollout has then constantly progressed in the majority of the considered countries with rapid achievements observed until the summer of 2021, when holidays were probably responsible for a slowdown, especially in countries like Italy, Romania, and Hungary. 

All the countries had access to COVID-19 vaccines at the same time in given amounts, calculated according to the size of their population. Most countries have defined priority, e.g., elderly aged 80 years or older, or healthcare workers, with several differences that could be explained with the general aim of decreasing mortality and pressure on hospitals and healthcare services. Many countries also adopted limitations for the administration of specific vaccines, in the presence of identified pathologies. Indeed, the use of the Vaxzevria and Janssen vaccines have been drastically limited starting from the spring 2021. On 11 March 2021, the European Medicines Agency (EMA) started reviewing the rare but serious side effects observed after some immunizations undertaken with Vaxzevria. The EMA expert group concluded that Vaxzevria was probably the cause of rare, serious symptoms, with blood clots combined with a low platelet count and bleeding. Notwithstanding, Vaxzevria remained a vaccine authorized by the drug regulatory authorities, but the EC decided to suspend any order for it beyond June 2021 when their contract ended [26]. Only Hungary has continued to use the Vaxzevria vaccine throughout the study period. The Vaxzevria incident underlines that good coordination and communication is essential by government, public health authorities, and medicines’ national and supranational agencies to determine a clear vaccination strategy, which can contribute to the reduction of hesitancy in the population.

The EC had a central role in supporting national authorities for the organization of a fast and accessible deployment of vaccines, and has issued guidance on the large-scale vaccination deployment [27]. One of the most important achievements was the creation of the “Inclusive Vaccine Alliance” countries, composed by Germany, France, Italy, and the Netherlands, who started in August 2020 negotiations with AstraZeneca company, and asked the EC to take over through an agreement signed on behalf of all member states [28]. This strategy aimed to secure for all European citizens high-quality, safe, effective, and affordable vaccines within 12 to 18 months from the agreement, and it was actually respected.

Several countries reported regional differences in vaccination uptake and faced challenges in reaching unvaccinated populations. In particular, difficulties were reported in increasing vaccination uptake in some population groups such as underserved/socially vulnerable individuals, and young people. Some countries introduced incentives to be vaccinated, while others required mandatory vaccination for healthcare workers. Additionally, most countries tried to facilitate access to vaccination through vaccination pop-up points without reservation, or sending vaccination appointment reminders [29].

From both a policy and a scientific viewpoint, it is fundamental to implement an effective monitoring system of vaccine administration in the population, both at national and international level, with the greatest achievable interoperability. Sharing these data, and examining and completing them with a proper communication strategy, could be useful to try to understand and counteract the reasons behind low vaccination uptake. Enough flexibility to change course should be then taken into account to quickly find different solutions if a particular strategy is found to be not effective.

Conversely, some of the strategies introduced for the COVID-19 vaccination campaign could be kept in place and used for other vaccinations, i.e., vaccination centers in hospitals, which can be used to vaccinate high-risk patients (patients with severe asthma or allergies, patients with many comorbidities, etc.) or large settings, like sports halls or similar, where some vaccinations (e.g., influenza) could be administered on large populations in a limited time. Genomic surveillance is also an essential tool for tracing the spread of SARS-CoV-2 at various scales, from individual transmission events to the intercontinental spread of the virus. In addition, it has had a central role in monitoring the evolution of SARS-CoV-2 and identifying new variants. Genomic surveillance also demonstrated the effectiveness of tracking local transmission events for informing public-health decision-making, and for adopting social-distancing measures to reduce viral spread [30]. Among the studied countries, this important tool for the tracing and quick detection of new variants was adopted with wide differences in Denmark, France, Italy, Spain, and Sweden, but not in Hungary, Germany, and Romania. This is a very important point to take into consideration as the spreading of new variants is growing and the rapid identification of them can allow a better response by countries, also considering that variants are becoming more and more contagious despite causing less severe disease.

The results of the present study are relevant for their implications in public health. They confirmed the hypothesis that heterogeneity and big differences in the planning and organization of a vaccination campaign during a pandemic are not positive for the aim of rapidly reaching high vaccination coverage rates. In supplying vaccines, a centralized agreement and exchange of information at a European/governmental level would also be probably useful for some aspects of the management of vaccination campaign, at least by those countries presenting similar health systems. However, it should also be important to learn from the positive experience of Israel’s rapid and efficient prioritization and administration of the vaccination [25]. Moreover, it is advisable to increase comparative research in this field as it can improve the knowledge of the main factors causing difficulties in the management and slow pacing of the rollout, as well as low coverage rates, and how to address them.

The main strengths of this study are represented by the different perspective taken to elaborate on the evolution of vaccination coverage, and the unavailability of similar studies on the same topic and regarding different European countries at the same time. Furthermore, we also tried to be as comprehensive as possible in our desk search. 

The limitations are represented by the time frame considered, due to the willingness to give a picture of the situation before the jeopardized utilization of the booster dose starting from Autumn 2021, the possible lack of information in the collected documents, even if institutional, and the potential differences between what was reported in the NIPs and accessed documents and what was really in place in the organization of vaccination campaign in each country. Furthermore, our analysis was not aimed at quantifying the impact of each specific aspect concerning the vaccination campaign on vaccination coverage. This issue, while interesting, would require a multilevel analysis that considers both group and individual aspects involved in vaccination uptake.

## 5. Conclusions

The planning and organization of the COVID-19 vaccination campaign can be recognized as one of the biggest challenges that countries faced during the COVID-19 pandemic. The eight countries included in this study showed different strategies put in place with some common issues, especially for prioritized categories of individuals to be vaccinated. The choices made highly reflect the organization of the countries’ health systems and could have been responsible for the different evolution of vaccination coverage. Best practices should be identified by the policy makers and all stakeholders in order to adapt the next COVID-19 vaccination campaigns, which are very likely to be implemented for autumn 2022. 

## Figures and Tables

**Figure 1 vaccines-10-01631-f001:**
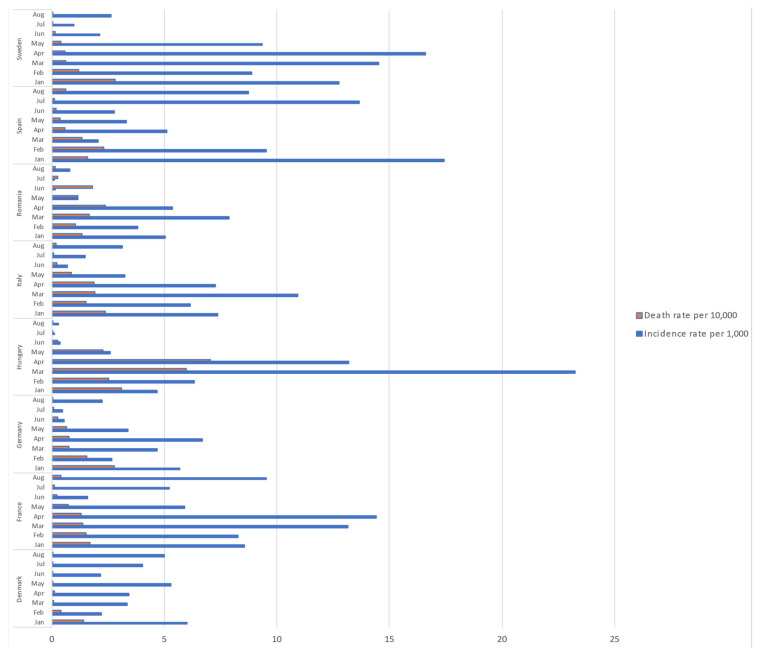
COVID-19 epidemiology in the eight selected countries. January–August 2021.

**Table 1 vaccines-10-01631-t001:** Full vaccination coverage by considered time points, percentage change (PC) and average percentage change (APC). January–August 2021.

Day	Time JPR	Denmark		France		Germany		Hungary		Italy		Romania		Spain		Sweden	
		Coverage	PC	Coverage	PC	Coverage	PC	Coverage	PC	Coverage	PC	Coverage	PC	Coverage	PC	Coverage	PC
01/01/2021		0.0		0.0		0.0		0.0		0.0		0.0		0.0		0.0	
15/01/2021		0.0		0.0		0.0		0.0		0.0		0.0		0.0		0.0	
01/02/2021	1	1.4	38.9	0.1	365.1	0.8	72.5	0.7	82.8	1.2	41.4	0.6	109.3	0.9	42.2	0.3	195.6
15/02/2021	2	2.6		1.1		1.8		1.4		2.3		2.4		2.3		1.4	
01/03/2021	3	2.8		2.5	31.0	2.7	38.0	2.6		2.5		3.2	31.8	2.7		2.7	29.9
15/03/2021	4	4.1		3.4		3.6		4.1		3.6		3.7		3.6		3.4	
01/04/2021	5	6.1		4.5		5.2		8.9		5.7		5.7		5.9		5.2	
15/04/2021	6	7.6		6.4		6.5		13.7	34.5	7.3		7.9		6.9		6.7	
01/05/2021	7	10.4		9.8		8.0		20.8		10.5		10.2		10.8		7.3	
15/05/2021	8	16.6		13.4		11.2		27.8		14.6		14.4		15.2		9.8	
01/06/2021	9	21.2	23.8	17.2		18.9		38.1		21.1	22.2	19.2		20.4		14.5	
15/06/2021	10	25.4		22.7		28.0		44.0	3.7	24.8		22.0	2.5	28.0		22.0	
01/07/2021	11	33.4		32.1		38.1		52.1		32.6		23.5		38.9	14.5	33.0	
15/07/2021	12	42.7		40.3		45.7	9.0	54.3		42.6		24.4		49.2		36.4	
01/08/2021	13	54.0		48.2		52.6		56.1		53.8		25.2		57.4		40.9	
15/08/2021	14	64.4		52.8		57.4		56.7		58.5		25.8		62.7		46.9	
01/09/2021	15	71.2		60.4		60.7		56.8		62.4		26.6		70.9		57.5	
Overall APC			32.2		57.0		35.5		36.8		32.9		28.7		33.6		36.8

## Data Availability

Not applicable.

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
