# Peer review of "Planning and Organization of the COVID-19 Vaccination Campaign: An Overview of Eight European Countries"

_vaccines, 2022, doi:10.3390/vaccines10101631_

Round 1

Reviewer 1 Report

There are minor revisions that I would like the authors to address and/or to consider.

1-      Please explain how accuracy of obtained data was assessed

2-      Please explain the method of selection of countries

3-      line 175; please amend:  80 years or odlers

4-      "Thanks to the joinpoint regression analysis" didn’t understand: Thanks??

5-      Please provide implications of the study

Author Response

5th September 2022

Dear Editor,

            We are submitting the revision of the manuscript no. vaccines-1865741, entitled “Planning and organization of the COVID-19 vaccination campaign: an overview of eight European countries” that we have submitted for publication on the Special Issue where we are Guest Editors for Vaccines.

            We have made some changes in all the sections, as per the Reviewer’s comments. All suggestions indicated by the Reviewers have been taken into account. The following are our replies.

#Reviewer 1

There are minor revisions that I would like the authors to address and/or to consider.

  1. Please explain how accuracy of obtained data was assessed.

Answer: Thanks for this request of clarification. A cross-check of information and data were not performed because only official/institutional sources were consulted. Nonetheless, we have better clarify in the limits of the study that we cannot rule out potential discrepancies between information collected by official documents and real practice.

  1. Please explain the method of selection of countries

Answer: The selection of countries was opportunistic but aimed to be representative of the diverse European regions and health systems. We have included a sentence to clarify it.

  1. line 175; please amend: 80 years or odlers

Answer: thanks for the warning. The typo was corrected.

  1. "Thanks to the joinpoint regression analysis" didn’t understand: Thanks??

Answer: thanks for the warning. The beginning of the sentence was rephrased as follows “Based on”.

  1. Please provide implications of the study.

Answer:  Thanks for your request. We have enclosed a new paragraph with implications.

Reviewer 2 Report

The paper presents the results of an analysis of national COVID-19 vaccination programs in 8 European countries during the January-August 2021 period. The research is interesting, but the relevance and significance of the results presented are limited. I recommend to the authors to take into account the following comments to increase the quality and interest of the paper.

1. Abstract. It is too generic. It does not informs about the research strategy and the quantitative data obtained.

2. Lines 70-72. The research objective is too generic. The regression analysis is not a descriptive method, and its use must be related with the research objectives. It is necessary to include the study period in the research's objectives.

3. Lines 78-83.  It is necessary to include adequate references.   

4. Table 1. This table is confusing. It should be organized by vaccination items and countries.

5. Line 110. The vaccination program information covered by the study is too short and does not cover the August 2021-August 2022 period. The analysis of this period is relevant because vaccination programs were implemented after COVID-19 vaccines were approved.

6. Lines 113-120. The methodology of the statistical analysis is not explained in sufficient detail. What type of statistical analysis was carried out? What type of model was used and how it was developed? What were the dependent and independent variables in the model/s?

7. Line 125. What is the effect on the results derived from differences concerning when COVID-19 NIP were issued?

8. The Results section presents a description of the information collected from different countries.    

9. The results presented on lines 216-229 are not explained in sufficient detail. The relevance of detecting different phases in different countries must be explained.

10. Did the study detected the factors influencing on the vaccination phases detected in the study? 

11. Lines 233-234. The discussion about the planning and organization of COVID-19 vaccination programs is based on the comparison of the information collected during the study period. Nevertheless, the implications derived from the study results are not presentred and discussed.    

12. The Discussion does not give an overview of the factors influencing on the COVID-19 vaccination coverage, appart from those assessed in the study. Nevertheless, the journal Vaccines has published many papers on the factors influencing on the COVID-19 vaccination coverage in several Special  Issues. What do the paper adds to the current knowledge on this theme?    

Author Response

5th September 2022

Dear Editor,

            We are submitting the revision of the manuscript no. vaccines-1865741, entitled “Planning and organization of the COVID-19 vaccination campaign: an overview of eight European countries” that we have submitted for publication on the Special Issue where we are Guest Editors for Vaccines.

            We have made some changes in all the sections, as per the Reviewer’s comments. All suggestions indicated by the Reviewers have been taken into account. The following are our replies.

#Reviewer 2

The paper presents the results of an analysis of national COVID-19 vaccination programs in 8 European countries during the January-August 2021 period. The research is interesting, but the relevance and significance of the results presented are limited. I recommend to the authors to take into account the following comments to increase the quality and interest of the paper.

  1. It is too generic. It does not informs about the research strategy and the quantitative data obtained.

Answer: Thanks for your request. We have modified the abstract in order to provide more information about the research strategy (desk research) and main results.

  1. Lines 70-72. The research objective is too generic. The regression analysis is not a descriptive method, and its use must be related with the research objectives. It is necessary to include the study period in the research's objectives.

Answer: thanks for the suggestion. We have rephrased the objective clarifying that we looked at the initial 8 months of the vaccination campaign and that, as secondary objective, we did also speculate on the relationship between planning and organization of the COVID-19 vaccination campaigns and vaccination coverage. Nevertheless, for the sake of clarity, we must point out that we did not investigate any quantitative association between a specific factor and the vaccination coverage within the join point regression model. A sentence has been included at the end of discussion to make it explicit and acknowledge it as a limitation of the study.

  1. Lines 78-83.  It is necessary to include adequate references.   

Answer: thanks for your request. All the references are already included and are listed in Table 1S.

  1. Table 1. This table is confusing. It should be organized by vaccination items and countries.

Answer: thank you for the comment. We tried to apply your suggestion about organizing the table by vaccination item but it resulted more complex and confusing than how it is now. Therefore, we decided to move the table to the supplementary material. Indeed, this table reports only the sources used for each country, so we think it would be more appropriate to present it as a supplementary.

  1. Line 110. The vaccination program information covered by the study is too short and does not cover the August 2021-August 2022 period. The analysis of this period is relevant because vaccination programs were implemented after COVID-19 vaccines were approved.

Answer: thanks for this comment. We are aware that we limited our analysis to the very beginning period of the vaccination campaign, but this was a deliberate choice. There were different reasons behind this choice. As already reported in limitations we wanted to avoid considering the period from Autumn 2021 onward because of the jeopardized utilization of the booster dose. Furthermore, from the end of 2021 onwards, children vaccination was also launched. Furthermore, the European Union set the goal of vaccinating 70% of the European population before the end of the summer and our analysis was aimed to catch this time period. This was the main reason behind our choice and has been included at the beginning of methods section.  

  1. Lines 113-120. The methodology of the statistical analysis is not explained in sufficient detail. What type of statistical analysis was carried out? What type of model was used and how it was developed? What were the dependent and independent variables in the model/s?

Answer: thanks for your request. We adopted a specific regression model, known as joinpoint regression (JPR) to identify and evaluate when/if changes in parameters (namely the proportion of population covered by the vaccination) occur along a time series. The JPR assumes that data can be divided into subsets each with their own unique linear trend.  From the analytical point of view, the joinpoint regression model for the observations (x1,y1),…,(xn,yn), where x1<x2<...<xn represents the time variable (say the 15-day time periods) and yi (i=1,…,n) is the response variable (say the vaccination coverage) can be written as:

yi= β01xi1(xi1)+...+δk(xik)+εi

where:

where ?0, ?1, ?1, … ?k are regression coefficients with τk-1<xik and τ1<...<τk are the joinpoints (with ?1, …, ?k being slope differences given time period ?) and ??∼?(0, ?2). The joinpoint model assumes linearity, and errors ?? are independent and normally distributed.

The joinpoint regression is different from other similar models, like piecewise regression, because it has the constraint of continuity at the change-point(s) and the choice of the number of joinpoint(s) and their locations is estimated within the model. A distinguishing characteristic of this model is that the minimum and the maximum number of joinpoints allowed is arbitrarily set before the analysis while the final number of joinpoint(s) is not fixed a priori by the researcher, but it is established based on a statistical criterion.

Some more information about dependent and independent variables and assumptions of the model has been included in the text.

  1. Line 125. What is the effect on the results derived from differences concerning when COVID-19 NIP were issued?

Answer: thanks for your request. As reported above and underneath our aim was not to assess the role of each different aspect of the vaccination campaign in respect to the trend of vaccination coverage. In fact, our main objective was to provide an overview of how the vaccination campaign was managed. A sentence has been included at the end of discussion to make it explicit and acknowledge it as a limitation of the study.

  1. The Results section presents a description of the information collected from different countries.    

Answer: no actions required.

  1. The results presented on lines 216-229 are not explained in sufficient detail. The relevance of detecting different phases in different countries must be explained.

Answer: Thanks for your comments. In order to clarify why we looked at changes in vaccination coverage trends through the join point analysis we have included a sentence in the methods section of the amended version. In respect to the presentation of the results of the join point analysis we believe that all the details were already provided, also considering that table 2 reports all data and results. Nonetheless, we are more than willing to make other amendments according to more specific requests/comments of the reviewer.

  1. Did the study detected the factors influencing on the vaccination phases detected in the study? 

Answer: thanks for your request. Unfortunately, the join point regression did not assess the impact of specific factors on vaccination coverage but only identified changes in trends. This is why we restricted ourselves to speculate on some aspects that could have influenced the trend. A sentence has been included at the end of discussion to make it explicit and acknowledge it as a limitation of the study.

  1. Lines 233-234. The discussion about the planning and organization of COVID-19 vaccination programs is based on the comparison of the information collected during the study period. Nevertheless, the implications derived from the study results are not presentred and discussed.   

Answer:  Thanks for your request. We have enclosed a new paragraph with implications.

  1. The Discussion does not give an overview of the factors influencing on the COVID-19 vaccination coverage, appart from those assessed in the study. Nevertheless, the journal Vaccines has published many papers on the factors influencing on the COVID-19 vaccination coverage in several Special  Issues. What do the paper adds to the current knowledge on this theme?    

Answer: Thanks for this comment. It is well known that vaccination acceptance and, indeed, coverage, is due to several reasons. Nonetheless, most of the literature is focused on the reasons behind vaccination hesitancy and identified individual and group factors as the most important aspects. The originality of our work lies in the fact that we did not want to focus on vaccine hesitancy but on the management of vaccination campaign that, anyway, could have an influence on vaccination acceptance. We did not find so much literature on this aspect. Running the following search on PubMed ("COVID-19"[Mesh] AND "Immunization Programs"[Mesh] AND "Organization and Administration"[Mesh]) we only found 73 papers and none of them provided an analysis like ours. We have included an Israel’s study in the bibliography and a sentence at the beginning of discussion to better capture this aspect. We remain available to receive further indications on selected papers that we could have missed.

Round 2

Reviewer 2 Report

The revised version of the paper has clarified several unclear aspects of the research carried out in the study, but the paper has still the following problems:

1. The authors have not justified in a sufficiently consistent way the reasons for limiting the research to the period January-August 2021.

2. The research strategy based a pointjoint regression analysis is not explained in sufficient detail. For example, the methodology section does not explain in sufficient detail how the regression analysis was used to detect differences in the management of vaccination programs.

3. Line 119-120. The methodology section indicates that the vaccination coverage used in the research “was calculated as the proportion of persons partially or fully vaccinated”.  Nevertheless, the definition used for the results presented on Table 1 seems to be “completed or partly vaccinated”. For example, For Spain the coverage indicated on Table 1 is 78.18%, which is mentioned for “at least one dose” on line 215.

4. The data for the vaccination coverage used in the analysis is different from the data now available at the web Ourworldindata for the compared countries. For example, the coverages on 1/1/21 for Denmark, France, Germany, Hungary, Italy, Romania, Spain and Sweden were not those presented on Table 1 but the following ones: 0.56%, 0%, 1.33%, 1.34%, 0.33%, 1.05%, 1.88% and 1.08%, respectively. The coverages on 15/8/21 for Denmark, France, Germany, Hungary, Italy, Romania, Spain and Sweden were not those presented on Table 1 but the following ones: 73.06%, 69.55%, 64.23%, 58.48%, 68.94%, 26.47%, 73.11% and 63.43%, respectively. Consequently, the analysis presented in the paper is not correct.

Author Response

12th September 2022

Dear Editor,

            We are resubmitting the revision of the manuscript no. vaccines-1865741, entitled “Planning and organization of the COVID-19 vaccination campaign: an overview of eight European countries” that we have submitted for publication on the Special Issue where we are Guest Editors for Vaccines.

            We have made some changes in all the sections, as per the Reviewer comments. All suggestions indicated by the reviewer have been taken into account. We are grateful to the reviewer for his/her precious comments which helped to improve the manuscript. The following are our replies.

#Reviewer 2

The revised version of the paper has clarified several unclear aspects of the research carried out in the study, but the paper has still the following problems:

  1. The authors have not justified in a sufficiently consistent way the reasons for limiting the research to the period January-August 2021.

Answer: As reported in the Materials and Methods section, the European Union set the goal of fully vaccinating 70% of the European population before the end of the summer 2021 and our analysis was aimed to catch this time period. Moreover, this was the same period that just preceded the authorisation for the booster dose, given in Autumn 2021 (https://www.ema.europa.eu/en/news/comirnaty-spikevax-ema-recommendations-extra-doses-boosters) that changed the framework, the coverage, the efficacy issues related to variants and the considerations about the management. These were the main reasons behind our choice of limiting our research to those first months.

  1. The research strategy based a pointjoint regression analysis is not explained in sufficient detail. For example, the methodology section does not explain in sufficient detail how the regression analysis was used to detect differences in the management of vaccination programs.

Answer: The main aim of joinpoint regression (JPR) is to  identify time point(s) where the trend significantly changes. JPR was not used to assess the impact of specific factors on vaccination coverage. We just speculated on some aspects that could have influenced the trend.

  1. Line 119-120. The methodology section indicates that the vaccination coverage used in the research “was calculated as the proportion of persons partially or fully vaccinated”. Nevertheless, the definition used for the results presented on Table 1 seems to be “completed or partly vaccinated”. For example, For Spain the coverage indicated on Table 1 is 78.18%, which is mentioned for “at least one dose” on line 215.

Answer: We modified the analysis and the text according to the goal of EU (70% of fully vaccinated people among the adult population before the end of the summer 2021). So, in the revised version we considered only indicators related to people fully vaccinated.

  1. The data for the vaccination coverage used in the analysis is different from the data now available at the web Ourworldindata for the compared countries. For example, the coverages on 1/1/21 for Denmark, France, Germany, Hungary, Italy, Romania, Spain and Sweden were not those presented on Table 1 but the following ones: 0.56%, 0%, 1.33%, 1.34%, 0.33%, 1.05%, 1.88% and 1.08%, respectively. The coverages on 15/8/21 for Denmark, France, Germany, Hungary, Italy, Romania, Spain and Sweden were not those presented on Table 1 but the following ones: 73.06%, 69.55%, 64.23%, 58.48%, 68.94%, 26.47%, 73.11% and 63.43%, respectively. Consequently, the analysis presented in the paper is not correct.

Answer: As we specified in the answer raised in the point 3, in the amended version we considered only the percentage of people “fully vaccinated”. We also updated the information, accessing the ourworldindata.org/covid-vaccinations database on 9th of September 2022.

Round 3

Reviewer 2 Report

The revised version of the paper has clarified several unclear aspects of the research carried out in the study, but the paper has still several critical problems. The main of them are: 

1) The data for the percenatges of vaccination coverager in different countries used in the analysis is different from that public available at the web Ourworldindata, thus making the analysis not correct.

2) The vaccination coverage used in the analysis was "completed vaccination", as it is indicated in the paper, but the coverage for "at least one dose." 
